# Gel-Free Tools for Quick and Simple Screening of Anti-Topoisomerase 1 Compounds

**DOI:** 10.3390/ph16050657

**Published:** 2023-04-27

**Authors:** Josephine Geertsen Keller, Kamilla Vandsø Petersen, Karol Mizielinski, Celine Thiesen, Lotte Bjergbæk, Rosa M. Reguera, Yolanda Pérez-Pertejo, Rafael Balaña-Fouce, Angela Trejo, Carme Masdeu, Concepcion Alonso, Birgitta R. Knudsen, Cinzia Tesauro

**Affiliations:** 1VPCIR Biosciences ApS, 8000 Aarhus C, Denmark; 2Department of Molecular Biology and Genetics, Aarhus University, 8000 Aarhus C, Denmark; 3Department of Biomedical Sciences, Faculty of Veterinary Medicine, University of León, 24071 León, Spain; 4Department of Organic Chemistry, Faculty of Pharmacy, University of Basque Country (UPV/EHU), 01006 Vitoria-Gasteiz, Spain

**Keywords:** human topoisomerase 1, *Mycobacterium smegmatis* topoisomerase 1, *Leishmania donovani* topoisomerase 1, monkeypox virus topoisomerase 1, enzyme activity, rolling circle amplification, drug screening

## Abstract

With the increasing need for effective compounds against cancer or pathogen-borne diseases, the development of new tools to investigate the enzymatic activity of biomarkers is necessary. Among these biomarkers are DNA topoisomerases, which are key enzymes that modify DNA and regulate DNA topology during cellular processes. Over the years, libraries of natural and synthetic small-molecule compounds have been extensively investigated as potential anti-cancer, anti-bacterial, or anti-parasitic drugs targeting topoisomerases. However, the current tools for measuring the potential inhibition of topoisomerase activity are time consuming and not easily adaptable outside specialized laboratories. Here, we present rolling circle amplification-based methods that provide fast and easy readouts for screening of compounds against type 1 topoisomerases. Specific assays for the investigation of the potential inhibition of eukaryotic, viral, or bacterial type 1 topoisomerase activity were developed, using human topoisomerase 1, *Leishmania donovani* topoisomerase 1, monkeypox virus topoisomerase 1, and *Mycobacterium smegmatis* topoisomerase 1 as model enzymes. The presented tools proved to be sensitive and directly quantitative, paving the way for new diagnostic and drug screening protocols in research and clinical settings.

## 1. Introduction

DNA modifying enzymes, such as DNA topoisomerases, play an essential role in DNA replication, transcription, chromosome segregation, and recombination [1,2,3], as they are responsible for the removal of superhelical tension in genomic DNA [1,4]. Both eukaryotic and prokaryotic DNA topoisomerases are targets of small-molecule compounds with potential effects against human cancers or pathogens, such as eukaryotic parasites or bacteria causing diseases [5,6,7,8,9,10,11]. Hence, the need for new tools to investigate the clinical relevance of potential drugs against topoisomerases is highly relevant. Notably, the eukaryotic type 1B topoisomerases (TOP1B) are of high interest due to their potential as targets of compounds that are effective against cancer or pathogen-borne diseases [9,10,12,13]. Human topoisomerase 1 (hTOP1) helps to maintain genomic DNA integrity during cellular processes [2] and its activity is increased in several cancer types [14,15,16]. Other TOP1B include the monkeypox virus TOP1 (mpxvTOP1), which is a conserved enzyme playing an essential role in the replication and spreading of the monkeypox virus [17,18], and the *Leishmania donovani* TOP1 (LdTOP1), that is an essential enzyme involved in the replication of the *L. donovani* parasite, the causative agent of the neglected tropical disease visceral leishmaniasis [19]. Moreover, the *Mycobacterium tuberculosis* TOP1 (mtTOP1), a type 1A topoisomerase (TOP1A) present in the bacteria responsible for tuberculosis, is a relevant target for small-molecule compounds with anti-pathogen potential [20].

Common to these topoisomerases, is that they remove the topological tension in genomic DNA during cellular processes such as replication and transcription. This is achieved by the enzyme introducing a transient single-stranded break in DNA, thereby generating the formation of a cleavage intermediate with the enzyme linked to the 3′-end (for the TOP1B enzymes) or to the 5′-end (for the TOP1A enzymes) of the nicked DNA. Subsequently, the nicked DNA is religated and the enzyme leaves the DNA intact [1,4]. This catalytic cycle can be targeted by small-molecule compounds acting either as TOP1 inhibitors or TOP1 poisons [21]. TOP1 inhibitors act by preventing the catalytic activity by inhibiting the DNA binding and/or cleavage, whereas TOP1 poisons act by prolonging the half-life of the enzyme–DNA cleavage complex. This can result in accumulation of TOP1-bound nicks in the genome, which ultimately can lead to cell death by collision with the DNA replication and transcription machinery. Reduced TOP1 activity levels are detrimental for single-cell eukaryotes such as *Trypanosoma brucei* and *L. major* [22,23,24], while TOP1 appears only to be essential during early developmental stages in higher eukaryotes and mammals [25,26]. Hence, TOP1 inhibitors are of particular interest as potential anti-parasitic drugs [10], while TOP1 poisons are of high interest as potential anti-cancer agents, since they convert TOP1 activity into a cell killer [27]. Examples of TOP1-targeting anti-cancer drugs include the water-soluble camptothecin (CPT) derivatives, topotecan, and irinotecan [28].

In order to screen libraries of natural and synthetic small-molecule compounds for potential TOP1-targeting properties, it is necessary to have access to assays that allow easy and fast investigation of the effect of these compounds on TOP1 activity. Several assays, including the relaxation assay [29], electrophoretic mobility shift assay (EMSA) [30,31], the DNA suicide cleavage-ligation assay [32,33], and the in vivo complex of enzymes (ICE) assay [34], have been developed. These assays can be used to assess the inhibitory effect of new compounds. However, these assays all rely on gel electrophoresis, which requires DNA intercalating agents, or they require special equipment and expertise. Moreover, most of these assays only perform optimally when using relatively large amounts of purified TOP1 enzyme. To enable specific detection of hTOP1 activity in small and even crude samples, rolling-circle-enhanced enzyme activity detection (REEAD), was previously developed [35]. This assay relies on the hTOP1-mediated circularization of a specific DNA substrate. The closed circles are then amplified by isothermal rolling circle amplification (RCA), generating 10^3^ tandem repeats, which subsequently can be visualized either by a fluorescent-based readout or a chemiluminescent-based readout.

TB is currently the second-leading infectious disease, with 1.6 million deaths in 2022 [36]. The number of new cases may potentially rise in the near future due to growing worldwide mobility. This emphasizes how critical it is to establish tools for rapid TB diagnosis and for the screening of candidate molecules for the treatment of TB, counteracting the emergent antibiotic resistance.

Taking the challenges in measuring the activities of TOP1A and TOP1B into account, this study presents the easy and fast REEAD assay as a drug screening tool for compounds against hTOP1 and LdTOP1. It also shows how this assay can be adapted to investigate the activity of mpxvTOP1, as a model for poxviruses, even in crude samples. Moreover, an alternative use of the previously developed assay for the detection of mtTOP1 activity [37] is presented, using the non-pathogen *M. smegmatis* TOP1 (MsTOP1) as a model enzyme. The assay is termed TB enzyme activity detection (TB-EAD), and this study demonstrates the drug screening potential of this assay.

## 2. Results

### 2.1. Detection of hTOP1, LdTOP1, and mpxvTOP1 Activities Using the REEAD Assay

TOP1 inhibitors are significant compounds for their anti-cancer, anti-bacterial, and anti-parasitic capacity [6,9,10,11,13,38]. Efficient and sensitive methods for the detection and quantification of their inhibitory capacity are therefore highly desirable, as traditional control systems are expensive and require skilled personnel. Here, an assay capable of detecting the activity and the drug response of TOP1B is presented. The principle of this assay is illustrated in Figure 1 and relies on the TOP1-mediated circularization of an open substrate (Figure 1A) with or without the presence of TOP1-targeting drugs. The hTOP1- and LdTOP1-specific substrates fold into a dumbbell shape containing two single-stranded loops and a double-stranded stem. The double-stranded stem contains an hTOP1/LdTOP1-preferred cleavage site and one of the single-stranded loops contains a sequence complementary to a surface-anchored primer suitable for RCA. The mpxvTOP1-specific substrate folds into a half-dumbbell shape, with one single-stranded loop containing a sequence complementary to the surface-anchored primer and a double-stranded region containing a mpxvTOP1-preferred cleavage site. Upon TOP1-mediated cleavage and ligation, the circularized substrates are hybridized to the surface-anchored primers, in well-defined rectangular areas (termed wells) of a glass slide. The hybridized circles act as a template for isothermal RCA mediated by Phi29 polymerase (Figure 1B(i)). During RCA, biotinylated nucleotides are incorporated into the rolling circle products (RCPs) (Figure 1B(ii)). This allows binding of horseradish peroxidase (HRP)-conjugated anti-biotin antibody and visualization of the generated RCPs using an enhanced chemiluminescence (ECL)-based readout (Figure 1B(iii)). The presented REEAD assay provides a sensitive detection of TOP1 activity, as one TOP1-mediated cleavage-ligation reaction generates a single RCP, thereby making the assay directly quantitative.

### 2.2. Using REEAD as a Drug Screening Tool for Drugs against hTOP1 and LdTOP1

To demonstrate the functionality of the REEAD assay as a drug screening tool, the activity of hTOP1 and LdTOP1 in the presence of TOP1-targeting compounds was measured. The two indenoisoquinoline derivatives, LMP400 and LMP744 (see the structure of the compounds in Appendix A), which are known hTOP1 [39] and LdTOP1 inhibitors [40] were used, and the hTOP1-targeting anti-cancer drug CPT was included as a control (as indicated in Figure 2). Twelve nanograms of recombinant purified hTOP1 or LdTOP1 (see Appendix A for protein purifications [41,42]) was incubated with the specific substrate in the presence of 80 µM of the compounds or 5% of the solvent DMSO. The generated circles were amplified by RCA in the presence of biotin-labeled nucleotides, and the HRP-conjugated anti-biotin antibody was subsequently bound to the biotinylated RCPs. Visualization was performed using the ECL-based readout, and Figure 2A,B show the results of these analyses. Figure 2A,B upper panels show representative images of the intensities of the biotinylated RCPs when visualized using ECL. As evident from the graphical depictions, and as expected from the literature [5,43], CPT inhibits both the hTOP1 (Figure 2A lower panel) and LdTOP1 (Figure 2B lower panel) activities significantly. LMP400 has a stronger inhibitory effect on LdTOP1 compared to hTOP1, while LPM744 seemed to be more potent against hTOP1 rather than LdTOP1 when compared to CPT. However, the scope of these experiments was not to directly compare the inhibitory effects of the two compounds, but merely to demonstrate the ability of the REEAD assay to be used as a screening tool for compounds that act against human, parasite, or other eukaryotic TOP1.

The observed inhibitory effect of the compounds was moreover demonstrated to be dose dependent (see Appendix A). The results presented are in line with the literature [5,43] and clearly demonstrate the drug screening ability of the REEAD assay.

### 2.3. The REEAD Assay Can Be Used to Measure Poxvirus TOP1B Activity

Poxviruses encapsidate a type 1B topoisomerase essential for viral growth. As a model for a poxvirus TOP1B, mpxvTOP1 was used. To screen for the activity of mpxvTOP1, a substrate that folds into a half-dumbbell shape, containing the specific cleavage sequence recognized by the poxviruses TOP1, as first identified in the vaccinia virus TOP1 [44], was designed (see Figure 1A). To validate the specificity of the mpxvTOP1 REEAD, the ability of hTOP1 and mpxvTOP1 to cleave/ligate the half-dumbbell-shaped substrate was compared. Recombinant purified mpxvTOP1 (0–50 ng) (see Appendix A for protein purification) or hTOP1 was incubated with the half-dumbbell substrate to generate closed circles that subsequently were amplified by RCA in the presence of biotin-labeled nucleotides. The biotinylated RCPs were again visualized using the ECL-based readout. The upper panel of Figure 3A shows representative images of the intensities of the biotinylated RCPs when measuring 0–50 ng of purified mpxvTOP1 or hTOP1, and the lower panel of Figure 3A shows a graphical depiction of the resulting quantifications.

Figure 3A shows that the signal intensity rises when analyzing a higher amount of purified mpxvTOP1, but not when examining increasing amounts of purified hTOP1. This result clearly demonstrates the specificity of the mpxvTOP1 REEAD assay. However, it should be noted that this assay is not specific for mpxvTOP1 compared to other poxvirus TOP1B, but can be used to detect all poxvirus TOP1 in purified form.

The REEAD assay has previously proven to be a powerful tool for the simple and fast detection of TOP1 activity in crude biological samples [15,45,46,47,48,49,50]. Next, it was investigated if the mpxvREEAD assay could be used to detect mpxvTOP1 in a crude extract. For this purpose, we performed a titration experiment, spiking in purified mpxvTOP1 in saliva, as a crude biological specimen. Using a fixed amount of saliva and increasing amount of purified mpxvTOP1, it was possible to detect mpxvTOP1 activity above background level, as indicated in Figure 3B. Two negative controls were included, one containing only the DNA substrate (“No enzyme, no saliva” in Figure 3B), as a measure of the possible nonspecific incorporation of biotin-labeled nucleotides during RCA of a non-circularized substrate, and one containing only saliva and purified mpxvTOP1 enzyme but no DNA substrate (“No substrate” in Figure 3B), as a measure of the potential nonspecific binding of the anti-biotin antibody to the wells. Both these controls gave intensities around or below the intensity observed when the reactions were performed in the absence of purified mpxvTOP1 (0 ng) with a fixed amount of saliva. The result presented in Figure 3B shows that the mpxvTOP1 REEAD can be used to detect mpxvTOP1 in a crude extract. Again, it is important to highlight that, due to the designed substrate characteristic, this assay cannot be used for diagnostic purposes as it does not allow for discrimination between TOP1B from different poxviruses.

### 2.4. Detection of MsTOP1 Activity Using the TB-EAD Assay

As mentioned, TOP1 inhibitors are relevant compounds for their anti-bacterial or anti-parasitic capacity. Here, the ability of the previously designed TB-EAD assay [37] to be used as a drug screening tool for new, more effective anti-TB drugs was investigated. TB-EAD is schematically depicted in Figure 4. The assay is based on the use of a single-stranded DNA substrate containing a strong TOP1 site (STS) [51], an RCA primer annealing sequence, and a probe annealing sequence. The substrate is hybridized to a surface-anchored RCA primer, and upon MsTOP1-mediated cleavage and ligation, the substrate is circularized (Figure 4A). This step occurs with or without the presence of the compound to be tested for inhibition ability. Starting from the surface-anchored primer, the closed circle is amplified by RCA mediated by Phi29 polymerase (Figure 4B(i)), generating a long tandem repeat product. By hybridization with fluorescently labeled probes (Figure 4B(ii)) to the generated RCPs, they can be visualized in a fluorescence microscope. The TB-EAD assay is highly sensitive and can detect MsTOP1 activity at a single molecule level, since each detected fluorescent spot in the microscope corresponds to a single catalytic cycle reaction of MsTOP1. Hence, the TB-EAD assay is also directly quantitative.

### 2.5. The TB-EAD Assay Can Be Used as a Drug Screening Tool for Drugs against MsTOP1

To demonstrate that the TB-EAD assay can be used as a drug screening tool for MsTOP1-targeting compounds, the activity of purified MsTOP1 was tested (see Appendix A for protein purification) in the presence of increasing concentrations of novel synthesized compounds, as indicated in Figure 5A. In the design of methods for the detection and quantification of this enzymatic inhibition, some heterocyclic compounds are key species due to their behavior against this target [52], as previously published [53,54,55,56]. Incidentally, it is important to mention that the recently optimized Povarov reaction [57,58] is a very suitable strategy, whose multicomponent version gives access to different families of heterocycles in a simple and fast manner. For this study, six compounds targeting MsTOP1 were synthesized (see Appendix A for the synthesis of the compounds and Appendix A for structural elucidation) to be used to test the drug screening ability of our REEAD-based assay. In total, 100 µM or 150 µM of the compounds, or 5% of the solvent DMSO, was added to the reaction mixtures during the circularization step of the substrate. Subsequently, the generated circles were amplified by RCA and visualized by hybridization to fluorescent probes. The number of signals was quantified, normalized to DMSO, and is graphically depicted in Figure 5A. Incubation with three compounds (**SF2**, **SF3** and **SF5**) and the starting reagent (**sulfadoxine 1**) showed a decrease in MsTOP1 activity already at the lowest dose, of 100 µM, whereas incubation with the compound **SF10** only showed a significant inhibition at the highest dose, of 150 µM. Incubation with the compound **HINSF1** did not result in any significant changes in the MsTOP1 activity.

One of the compounds showing the highest inhibitory effect on MsTOP1 activity was also tested against hTOP1 using the REEAD assay, shown in Figure 1. This was performed to show the specificity of the compounds against MsTOP1 alone. Indeed, when screening for new compounds against a specific TOP1 target, it is necessary to consider the specificity and potential off-target effects. In this case, having the possibility of testing both the human and bacterial enzyme, it was possible to assess if the tested compound could function as an anti-bacterial drug. Increasing concentrations of **SF5** were added to the circularization reaction of hTOP1, as indicated in Figure 5B. The circles were amplified by RCA and the resulting RCPs were visualized using the ECL-based readout. As evident from the graphical depiction, the two highest concentrations of **SF5** used show some inhibitory effect on hTOP1 activity, but to a lower extent compared to the effect on MsTOP1 activity. This demonstrated that **SF5** shows a small off-target effect. The results presented clearly demonstrate the ability of the TB-EAD assay to be used as a drug screening tool for compounds targeting MsTOP1.

## 3. Discussion and Conclusions

The ongoing demand for the discovery of novel small-molecule compounds with anti-cancer or anti-pathogen effects requires the development of easily accessible tools for measuring the interaction between a vast number of natural or synthesized new drugs and their potential cellular targets. Due to their key role in the maintenance of the genome integrity, topoisomerases have been targets of investigations of inhibitor compounds for the treatment of several types of cancers [12,13] since their discovery. In addition, topoisomerase inhibitors have also been investigated for their potential as anti-bacterial or anti-parasite infections agents [8,10]. These studies typically involve in vitro experiments using cell cultures along with different gel-based assays, which are labor and time consuming and require specialized equipment [29,30,31,32,33,34]. The presented study shows the advantages of RCA-based assays for the quick and simple screening of libraries of small-molecule compounds with potential effects as TOP1 inhibitors. As a proof of principle, we presented the REEAD-based assay for the measurement of the inhibitory effect of small-molecule compounds against eukaryotic and poxvirus TOP1B, using commercially available drugs with well-known effects, and using human, leishmanial, and monkeypox virus TOP1 as model enzymes. The presented tool will help with pre-screening of a large number of compounds to measure their potential inhibitory ability. More assays can be combined to test for target specificity, while other tools can be applied to investigate the mechanisms of the compounds in detail [59] and identify TOP1 inhibitors versus TOP1 poisons.

We demonstrated that the REEAD assay, with a novel developed ECL-based readout, is a valid alternative to the relaxation and radiolabeled oligonucleotide-based assays when rapid screening of a panel of a compounds is required. Moreover, the assay is easy and adaptable to every laboratory setting and is directly quantitative.

One of the most challenging diseases to fight, and with a high and growing spread around the globe, is TB. Most cases of TB can be treated with a combination of antibiotics [60], but some strains of TB have developed resistance and more resistant strains are emerging [61]. This highlights the importance of developing more TB drug discovery pipelines. We have previously developed an assay that can be used as a diagnostic tool for the identification of TB infection directly from saliva from patients [37]. Here, we demonstrated that the same tool is particularly useful also for the identification of novel anti-TB drugs, using mtTOP1 as the target.

We believe that the presented results provide solid tools that, in principle, might significantly shorten the time required to move from the identification of a possible inhibitor into a clinical trial.

## 4. Materials and Methods

### 4.1. Reagents

All chemicals were purchased from Sigma Aldrich, Søborg, Denmark.

### 4.2. DNA Oligonucleotides

DNA oligonucleotides were synthesized by LGC Biosearch Technologies, Lystrup Denmark. The sequences were as follow:5′amine REEAD primer: 5′-/5AmMC6/CCAACCAACCAACCAAGGAGCCAAACATGTGCATTGAGGhTOP1/LdTOP1 dumbbell substrate: 5′-AGAAAAATTTTTAAAAAAACTGTGAAGATCGCTTATTTTTTTAAAAATTTTTCTAAGTCTTTTAGATCCCTCAATGCACATGTTTGGCTCCGATCTAAAAGACTTAGAmpxvTOP1 half-dumbbell substrate: 5′-ATTGTATCGGAATAAGGGCGACAGACTCACTGTGAAGATCGCTTATCCTCAATGCACATGTTTGGCTCCGAGTCTGTCGCCCTTATTMsTOP1 substrate: 5′-CAGTGAGCGAGCTTCCGCTTGACATCCCATATCTCTACTGTGAAGATCGCTTATTCTCTCCTCAATGCACATGTTTGGCTCCTCTCTGAGCTTCCGCTFluorescent probe: 5′-FAM-CCTCAATGCACATGTTTGGCTCC

### 4.3. REEAD

#### 4.3.1. Preparation of Slides

A custom-designed silicone grid (Grace-bio lab, Bend, Oregon, USA), termed the Wellmaker, was attached to a CodeLink Activated HD slide (Surmodics, Saint Paul, Minnesota) and the slides were coupled with 10 µM of the 5′amine REEAD primer in 300 mM Na_3_PO_4_, pH 8. The slide was incubated overnight in a humidity chamber with saturated NaCl. The slide was subsequently blocked in 50 mM Tris, 50 mM Tris-HCl, and 50 mM ethanolamine pH 9, for 30 min at 50 °C and washed in 4× SSC, 0.1% SDS for 30 min at 50 °C.

#### 4.3.2. Circularization for Drug Screening

Circularization of the hTOP1- or LdTOP1-specific substrates was carried out by incubating purified hTOP1 or LdTOP1 with 0.1 µM of the specific substrate in the presence of 80 µM of the drugs to be tested (as indicated in the figure legend of Figure 2A,B) in a buffer containing 10 mM Tris-HCl pH 7.5, 5 mM EDTA, and 50 mM NaCl for 30 s for hTOP1 and 1 min for LdTOP1, at 37 °C. The reaction was stopped by the addition of 0.5% SDS. Subsequently, the circles were hybridized on the REEAD primer coupled slides for 1 h at 37 °C. The slide was then washed in wash buffer 1 (100 mM Tris-HCl pH 7.5, 150 mM NaCl, 0.3% SDS) for 1 min at room temperature, wash buffer 2 (100 mM Tris-HCl pH 7.5, 150 mM NaCl, 0.05% Tween20) for 1 min at room temperature, and finally dehydrated for 1 min in 70% EtOH.

#### 4.3.3. Circularization for mpxvTOP1-Specificity Test

Specific circularization of the mpxvTOP1 substrate was carried out by incubating either purified mpxvTOP1 or hTOP1 with 0.1 µM of the mpxvTOP1-specific substrate in a buffer containing 10 mM Tris-HCl pH 7.5, 5 mM EDTA, and 50 mM NaCl, for 1 h at 37 °C. The reaction was stopped by increasing the NaCl concentration to 250 mM. The hybridization of the circles to the slides and washing of the slides were performed as described above.

Alternatively, mpxvTOP1-mediated circularization of the specific substrate was carried out in the presence of a fixed amount of saliva. Here, increasing amounts of purified mpxvTOP1 (as indicated in the figure legend of Figure 3) were incubated with 0.5 µM of the specific substrate in the presence of 2 µL of saliva. The circularization was performed as described above.

#### 4.3.4. RCA and Detection of RCPs

RCA was performed in 1× Phi29 buffer (50 mM Tris-HCl pH 7.5, 10 mM MgCl_2_, 10 mM (NH_4_)_2_SO_4_, 4 mM DTT) supplemented with 0.2 µg BSA, 100 µM dATP, 100 µM dTTP, 100 µM dGTP, 90 µM dCTP, 10 µM biotin-dCTP, and 1 Unit Phi29 polymerase. The reaction was carried out for 2 h at 37 °C in a humidity chamber. The slide was then washed with wash buffers 1 and 2 and 70% EtOH, as previously. Subsequently, the slide was blocked in 1× TBST (20 mM Tris-HCl pH 9, 150 mM NaCl, 0.05% Tween20 pH 9) supplemented with 5% skimmed dry milk and 5% BSA for 1 h at room temperature, followed by incubation with 1:300 HRP-conjugated anti-biotin antibody in 1× TBST supplemented with 5% skimmed dry milk and 5% BSA, for 1 h at room temperature. The slide was washed 3 × 3 min in 1× TBST before addition of 2 µL of 1:1 ECL mixture to allow chemiluminescence readout using a CCD camera.

### 4.4. TB-EAD

#### 4.4.1. Preparation of slides

Slides were prepared as in REEAD (see step Section 4.3.1).

#### 4.4.2. Circularization

Ten picomoles of the MsTOP1 substrate, in 10 mM Tris-HCl pH 7.5, 1 mM EDTA, and 200 mM NaCl, was hybridized to the REEAD primer on the slides, for 1 h at 37 °C in a humidity chamber. The slides were washed for 1 min in wash buffer 1, 1 min in wash buffer 2, and 1 min in 70% EtOH, as described previously. Then, 100 ng of purified MsTOP1 was added to the slides in a buffer containing 10 mM Tris-HCl pH 7.5, 10 mM MgCl_2_, 10 mM MnCl_2_, 200 mM NaCl, 1 mM DTT, 0.1% Tween20, 100 µg/mL BSA in the presence of DMSO, or 100 µM or 150 µM of each of the drugs, as indicated in the figure legend of Figure 5. The reactions were incubated for 90 minutest 37 °C in a humidity chamber before the slides were washed in wash buffers 1 and 2 and 70% EtOH, as previously.

#### 4.4.3. RCA and Detection of RCPs

RCA was performed in 1× Phi29 buffer (50 mM Tris-HCl pH 7.5, 10 mM MgCl_2_, 10 mM (NH_4_)_2_SO_4_, 4 mM DTT) supplemented with 0.2 µg BSA, 1 mM dNTP, and 1 unit of Phi29 polymerase. The reaction was carried out for 1 h at 37 °C in a humidity chamber, followed by wash in wash buffers 1 and 2 and 70% EtOH, as previously. Two picomoles of the fluorescent probe were added to the slides in 2× SSC, 20% formamide, and 5% glycerol for 30 min in a humidity chamber at 37 °C. Subsequently, the slides were washed in wash buffer 1 for 10 min, wash buffer 2 for 5 min, and 1 min in 70% EtOH. The slides were mounted with Vectashield (Vector laboratories, Burlington, ON, Canada) and a cover glass, and analyzed in a fluorescence microscope with a 60× magnification objective. Twelve images were acquired for each sample and quantified using Image J Fiji.

### 4.5. Statistical Analysis

Data were analyzed using the GraphPad Prism software and expressed as mean +/− standard error of the mean (SEM). Statistical significance was assessed using one-way ANOVA test applying Brown–Forsythe and Welch correction.

## Figures and Tables

**Figure 1 pharmaceuticals-16-00657-f001:**
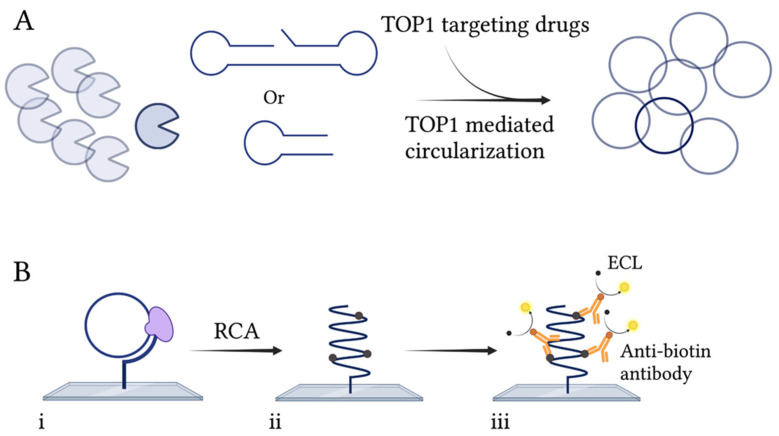
Schematic illustration of the REEAD assay. (**A**) TOP1 will cleave and ligate a specific substrate, thereby generating closed circular substrates. (**B**) (**i**) The closed circles are hybridized to a primer attached to a glass surface and can be amplified by RCA using the Phi29 polymerase (purple). (**ii**) In the presence of biotin-coupled nucleotides, the generated RCPs will be biotinylated, allowing the binding of HRP-conjugated anti-biotin antibodies. (**iii**) Visualization of the RCPs is carried out by addition of the ECL components, luminol and H_2_O_2_, that are converted into detectable light by HRP. Created with BioRender.com.

**Figure 2 pharmaceuticals-16-00657-f002:**
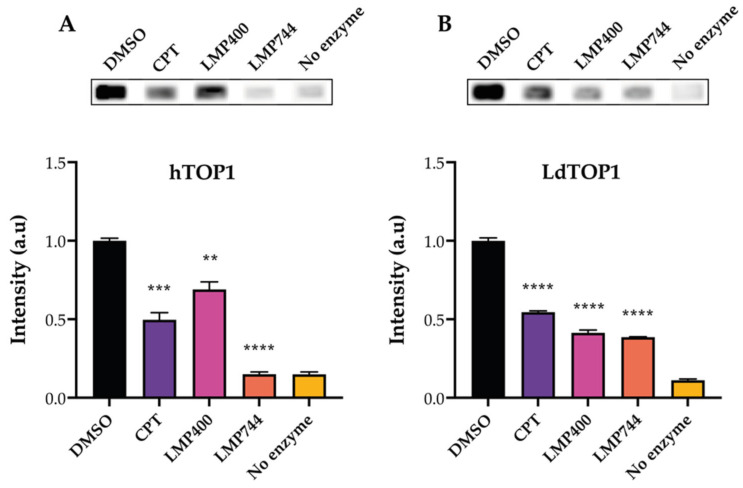
REEAD as a drug screening tool. (**A**) Top panel: image obtained after analyzing activity of purified hTOP1 in the presence of 5% of the solvent DMSO, or 80 µM of either of the compounds CPT, LPM400, or LMP744, using REEAD with the ECL-based readout. Lower panel: graphical depiction of the quantification of the results obtained when analyzing hTOP1 activity in the presence of 80 µM of the compounds indicated on the figure. A negative control, without enzyme, was included. Plotted data are normalized to the intensity obtained when measuring hTOP1 activity in the presence of DMSO and represent average +/− standard error of the mean (SEM) from six independent experiments. One-way ANOVA with Brown–Forsythe and Welch correction. Asterisks indicate significant difference compared to DMSO, ** *p* < 0.005; *** *p* < 0.0005; **** *p* < 0.0001. a.u: arbitrary units. (**B**) same as (**A**), except that LdTOP1 was used instead of hTOP1. Plotted data represent average +/− SEM from three independent experiments. One-way ANOVA with Brown–Forsythe and Welch correction, **** *p* < 0.0001. a.u: arbitrary units.

**Figure 3 pharmaceuticals-16-00657-f003:**
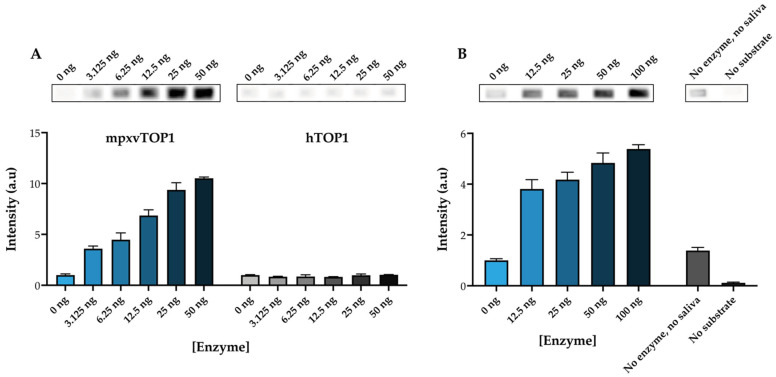
mpxvTOP1 REEAD assay. (**A**) Top panels: image obtained after analyzing the activity of 0–50 ng of purified mpxvTOP1 and hTOP1 using mpxvREEAD with ECL-based readout. Lower panel: graphical quantification of the results obtained when analyzing the activity of 0–50 ng of purified mpxvTOP1 or hTOP1 using the mpxvREEAD. Plotted data are normalized to the intensity obtained when analyzing 0 ng of purified enzyme and represent average +/− SEM from three independent experiments. a.u: arbitrary units. (**B**) Top panels: image obtained after analyzing the activity of 0–50 ng purified mpxvTOP1 containing a fixed amount of saliva spike using mpxvREEAD with ECL-based readout. Lower panel: graphical quantification of the results obtained when analyzing the activity of 0–50 ng of purified mpxvTOP1 containing a fixed amount of saliva spike using mpxvREEAD. Plotted data are normalized to the intensity obtained when analyzing 0 ng of purified enzyme and represent average +/− SEM from three independent experiments. a.u: arbitrary units.

**Figure 4 pharmaceuticals-16-00657-f004:**
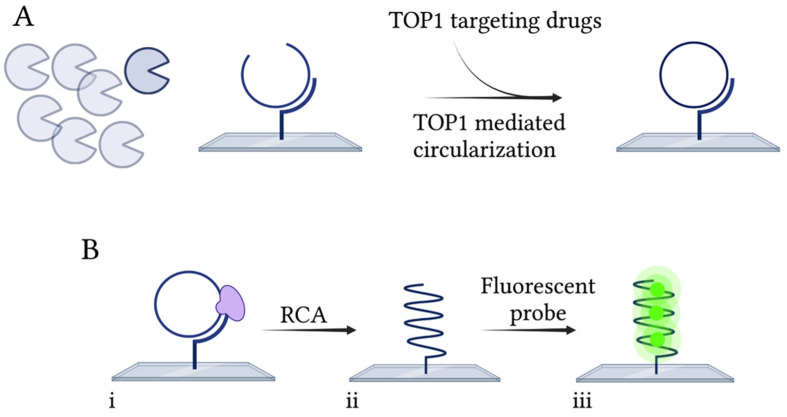
Schematic illustration of the TB-EAD assay. (**A**) The substrate is hybridized to a glass surface containing a complementary primer. MsTOP1 will then cleave and ligate the substrate, thereby making a closed circle. (**B**) (**i**) Following circularization, the circle is amplified by RCA generating tandem repeats (**ii**). (**iii**) Fluorescent probes are then hybridized to the RCPs, enabling visualization of the products in a fluorescence microscope. Created with BioRender.com.

**Figure 5 pharmaceuticals-16-00657-f005:**
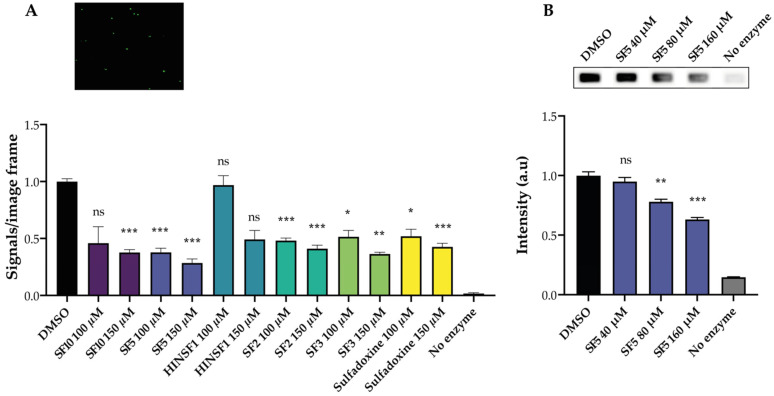
TB-EAD as a drug screening tool. (**A**) Top panel: representative microscopic image when analyzing purified mtTOP1 using the TB-EAD assay. Lower panel: graphical quantification of the results obtained when analyzing purified MsTOP1 activity using the TB-EAD assay in the presence of 100–150 µM of the compounds **SF10**, **SF5**, **HINSF1**, **SF2**, **SF3**, **sulfadoxine 1**, or 5% of the solvent DMSO. A negative control without enzyme was included. Plotted data are normalized to the number of signals obtained when measuring purified MsTOP1 activity in the presence of DMSO, and represent average +/− SEM from three independent experiments. One-way ANOVA with Brown–Forsythe and Welch correction. Asterisks indicate significant difference compared to DMSO, ns: non-significant; * *p* < 0.05; ** *p* < 0.005; *** *p* < 0.0005. (**B**) Top panel: image obtained after analyzing the activity of purified hTOP1 in the presence of 40–160 µM of the compound **SF5**, or 5% of the solvent DMSO, as indicated using the REEAD assay with ECL-based readout. Lower panel: graphical quantification of the results obtained when analyzing hTOP1 activity in the presence of 40–160 µM of the compound **SF5**. A negative control without enzyme was included. Plotted data are normalized to the intensity obtained when measuring hTOP1 activity in the presence of DMSO and represent average +/− SEM from three independent experiments. One-way ANOVA with Brown–Forsythe and Welch correction. * indicates significant difference compared to DMSO, ns: non-significant; ** *p* < 0.005; *** *p* < 0.0005.

## Data Availability

Data is contained within the article; raw data are available upon request from the corresponding author.

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
