# Peer review of "Gel-Free Tools for Quick and Simple Screening of Anti-Topoisomerase 1 Compounds"

_pharmaceuticals, 2023, doi:10.3390/ph16050657_

Round 1

Reviewer 1 Report

The authors have reported the use of already developed assay for the detection of mtTOP1 activity is presented, using the non-pathogen M. smegmatis TOP1 (MsTOP1) as a model enzyme.

The manuscript suffers from poor English language. Overall, it needs careful revision. Punctuation has not been used carefully.

Abstract needs to be revised carefully and it should briefly highlight significant results.

Literature gap has been sufficiently addressed, however, authors should highlight the merits of these methods in comparison to already reported methods. To support this, authors are encouraged to provide a table of comparison with already established assays. 

"Results and Conclusion" heading is probably against the journal's format and needs to be separated.

"Materials and Methods" should be provided with related references, where applicable.

Figure S3 should be part of manuscript not supplementary files. Similarly, synthesis of these compounds should be made part of main document. 

NMR spectra may be provided for synthesized compounds.

Author Response

Dear editor, we have written our answers to the referee in red

Point 1) The authors have reported the use of already developed assay for the detection of mtTOP1 activity is presented, using the non-pathogen M. smegmatis TOP1 (MsTOP1) as a model enzyme. The manuscript suffers from poor English language. Overall, it needs careful revision. Punctuation has not been used carefully. We thank the reviewer for the comment.

Response 1) We have carefully revised the entire manuscript and checked for spelling and punctuation.

Point 2) Abstract needs to be revised carefully and it should briefly highlight significant results.

Response 2) We have revised the abstract and highlighted the results, within the space allowed for the abstract in the journal guidelines.

Point 3) Literature gap has been sufficiently addressed, however, authors should highlight the merits of these methods in comparison to already reported methods. To support this, authors are encouraged to provide a table of comparison with already established assays. 

Response 3) We thank the reviewer for addressing the need to compare the described tools with the state of the art. We have mentioned the most significant findings in the literature. However, for a comprehensive comparison with all the research diagnostic/drug screening findings that are in line with the presented tools, we believe that a review publication format would be needed instead of a research article.

Point 4) "Results and Conclusion" heading is probably against the journal's format and needs to be separated.

Response 4) We have followed the guidelines provided by the journal, e.g that it is possible to have a section Discussion and Conclusion.

Point 5) "Materials and Methods" should be provided with related references, where applicable.

Response 5) We have included references where needed, e.g in the materials and methods section of supplementary informations.

Point 6) Figure S3 should be part of manuscript not supplementary files. Similarly, synthesis of these compounds should be made part of main document.

Response 6) For the length provided by the journal guidelines the synthesis of the compounds can be part of the supplementary informations.

Point 7) NMR spectra may be provided for synthesized compounds.

Response 7) NMR spectra are now included in supplementary informations  for the compounds not previously published.

Reviewer 2 Report

The manuscript pharmaceuticals-2266619 devoted the actual field of medicinal chemistry, namely development new tools for investigating the enzymatic activity of biomarkers and can be interested to the specialists working in this field. The author’s opinion is clear and based on a good experimenytal data. I am personally impressed by the structure of the article, the systematization of scientific data and the sequence of its presentation. The paper fit the Journal scope and formal requirements. However, it needs minor revision before publication.

To improve the quality and perception of the manuscript I would suggest paying attention to following comments:

1.     Considering the specifics of the journal, it would be good to provide the structure of the studied LMP400 and LMP744.

2.     The style of references  should be changed. In some cases, there are from 6 to 7 sources after one sentence (for example, lines 41, 318, etc). This is unacceptable for publications in high-rated journals. Instead such references, it would be better to make a cross-reference discussion.

My decision is m
inor revision.

Author Response

Dear Editor,

our responses are in red.

Point 1) Considering the specifics of the journal, it would be good to provide the structure of the studied LMP400 and LMP744.

Response 1) We thank the reviewer for this comment. The structure is now included in the supplementary information S1.

Point 2) The style of references should be changed.  In some cases, there are from 6 to 7 sources after one sentence (for example, lines 41, 318, etc). This is unacceptable for publications in high-rated journals. Instead such references, it would be better to make a cross-reference discussion.

Response 2)  We thank the reviewer for the comment. We have used the journal citation style. and it would indeed be valuable with a cross-reference discussion. However, such a discussion would only make the section longer and with a more “review” publication format style. This is out of the scope of the article, which is a research manuscript to present new method tools.